

# Seismogenic depth and seismic coupling estimation in the transition zone between Alps, Dinarides and Pannonian Basin for the new Slovenian seismic hazard model

Polona Zupančič[1], Barbara Šket Motnikar[1], Michele M. C. Carafa[3], Petra Jamšek Rupnik[2], Mladen Živčić[1], Vanja Kastelic[3], Gregor Rajh[1], Martina Čarman[1], Jure Atanackov[2], Andrej Gosar[1,4]

*Correspondence to*: Polona Zupančič (polona.zupancic@gov.si)

[1]Slovenian Environment Agency, Ljubljana, Slovenia
[2]Geological Survey of Slovenia, Ljubljana, Slovenia
[3]INGV
[4]University of Ljubljana, Faculty of Natural Sciences and Engineering, Ljubljana, Slovenia

**Abstract.** The seismogenic depth and seismic coupling are important inputs into seismic hazard estimates. Although the importance of seismic coupling is often overlooked, it significantly impacts seismic hazard results. We present an estimation of upper and lower seismogenic depth and hypocentral depth and seismic coupling in the transition zone between the Alps, Dinarides and Pannonian Basin, characterized by complex deformation pattern, highly variable crustal thickness, and moderate seismic hazard, supporting the development of the 2021 seismic hazard model of Slovenia. We estimated the lower seismogenic depth using seismological and geological data and compared them. The seismological estimate was based on two regional earthquake catalogues prepared for this study. In the area source model, estimates of lower seismogenic depth from seismological data are deeper or equal to the ones derived from geological data, except in one case. In the fault source model, we analyzed each fault individually and chose seismological lower depth estimates in 12 among 89 faults as more representative. The seismogenic thickness for each individual fault source was determined for seismic coupling determination. The seismic coupling was assessed by two approaches, i.e. we chose the most trusted value from the literature, and the value determined for each fault individually by using the approach based on the updated regional fault and earthquake datasets. The final estimate of seismic coupling ranges from 0.77 to 0.38. We compared the tectonic moment rate based on long-term slip rate using different values of seismic coupling with the seismic moment rate obtained from the earthquake catalogue. The analysis is done for the whole area, as well as for the individual area zones. The analysis of N-S components of estimated slip for the largest faults in the area of west Slovenia shows that the regional geologic and geodetic shortening rates are comparable. The total activity rate of three global seismic source models is compared, which gives up to a 10 % difference. Our results contribute to a better understanding of the seismic activity in the region and the approach for seismic coupling estimation can be applied in other similar regions.

**Keywords:** probabilistic seismic hazard assessment, parametrisation, seismic coupling, coupled thickness, seismogenic depth, Slovenia.



## 1. INTRODUCTION

Slovenia is located at the junction of the Alps, the Dinarides, and the Pannonian Basin, where different tectonic units contribute
to a complex deformation pattern (Schmid et al., 2020; Atanackov et al., 2021) (Fig. 1). Due to the Adria-Europe collision, the
area is under N-S compression and experiences counterclockwise rotation, leading to activation of strike slip and reverse
faulting in the area (e.g., Vrabec and Fodor, 2006; Placer et al., 2010; Poljak et al., 2010; Weber et al., 2010; Moulin et al.,
2016; Atanackov et al., 2021; Grützner et al., 2021). The most recent damaging earthquakes occurred on the strike-slip Ravne
Fault in 1998 and 2004 with $m_w$ 5.6 and $m_w$5.2, respectively (Bajc et al., 2001; Kastelic et al., 2008; Gosar et al., 2019a, 2019b).
Historical earthquakes with estimated moment magnitude $m_w$ of up to 6.5 imply the region experiences moderate seismicity
and has a moderate seismic hazard (Shedlock et al., 2000). Recently, the seismic hazard of the region has been assessed using
a modern approach that takes into account knowledge of active faults and seismicity (Šket Motnikar et al., 2022; Atanackov
et al., 2022). Understanding crustal structure is critical for developing a seismic hazard model, but available high-resolution
data covering the whole studied area are sparse.
The crustal structure of the studied region was investigated by several authors either along profiles (e.g., Brückl et al., 2007),
only with a few seismic stations (Michelini et al., 1998), or focused more on neighbouring (Najafabadi et al., 2022) or smaller
regions (Bressan et al., 2009). Guidarelli et al. (2017) resolved the S-velocity structure of the crust and uppermost mantle and
covered most of the territory of Slovenia and its surroundings. On the other hand, Kapuralić et al. (2019) computed a 3-D P-
wave velocity model at the junction of the Dinarides and the Pannonian basin from local earthquake tomography (LET).
Neither of these inversions was able to resolve small-scale anomalies in the upper crust and inverted only for one type of
velocity, which can bias either the velocity model or earthquake hypocentres. As we are using mainly seismological data in
seismic hazard analysis and sometimes infer some parameters from geophysical data, this can affect the results. Recently, Rajh
et al. (2022) inverted jointly for hypocentre parameters and 1-D P- and S-wave velocity models with station corrections. A
joint inversion for 3-D P- and S-wave velocity models from LET (Rajh, 2022) improved the earthquake locations even further
and provided additional insight into the upper crustal structure. Both studies showed that the hypocentres located in previous
studies were located too deep. Depth to the Mohorovičić discontinuity (Moho) in the studied area has been constrained by
several different studies (e.g., Brückl et al., 2007; Grad et al., 2009; Stipčević et al., 2011; Guidarelli et al., 2017; Kapuralić et
al., 2019; Stipčević et al., 2020). It ranges from about 38 to 45 km under the External Dinarides, becomes deeper towards the
Alps, and shallows to about 30 and 25 km in the Adriatic foreland and the Pannonian Basin, respectively. Seismogenic depth
analysis for earthquakes in Slovenia (Rajh et al., 2017; Rajh and Gosar, 2018) identified at least two distinct areas with
relatively shallow or deep earthquake foci. Upon close examination and as suggested already by several papers (Stipčević et
al., 2020; Rajh et al., 2022), one can observe that the depth distribution of earthquakes approximately follows the varying
Moho topography. This has an impact on the seismogenic depth of seismic sources.



**Figure 1. The seismotectonic setting of the study area: tectonic units are simplified from Schmid et al. (2020), active faults are summarized after Atanackov et al. (2021) and Poli and Zanferrari (2018), seismicity for $m_w > 5.0$ is from ARSO KPN2018 catalogue (1279-2018), focal mechanisms for $m_w > 5.0$ is from Bajc et al. (2001), Pondrelli et al. (2006), Kastelic (2008), and Herak et al. (2021).**

In 2021, a new Slovenian seismic hazard model and map were developed (Šket Motnikar et al., 2022), as a result of a seven-year joint project of the Slovenian Environment Agency (ARSO) and the Geological Survey of Slovenia (GeoZS). The following year, the hazard map became part of Slovenian legislation for earthquake-resistant design. During the two-year transition period, in addition to the new Slovenian seismic hazard map, the previous hazard map from 2001 (ARSO, 2001; Lapajne et al., 2003) is still officially valid. We applied probabilistic seismic hazard assessment (PSHA), as introduced by

Cornell (1968), which was later improved (e.g. Reiter, 1990; Baker et al., 2021). For the calculation of the new Slovenian seismic hazard map, the computer program OpenQuake was used (GEM, 2022).

In parallel to the development of the Slovenian hazard model, ESHM20 project, aiming to update the European seismic hazard model, was also underway (Danciu et al., 2021). This not only enabled us to exchange data and models, but we have also benefited from many discussions at regional meetings.






The new seismic hazard model includes three seismic source models (Fig. 2): a fault seismic source model (F), an area seismic source model (A), and a smoothed seismicity represented as a point seismic source model (P). The overview and results of the 2021 seismic hazard model for Slovenia, together with a description of the three seismic source models, their incorporation in seismic hazard calculation, and the estimation of most seismic source parameters are in detail explained in Šket Motnikar et al. (2022). The parameterization tables for area seismic sources (AS) and fault seismic sources (FS), as well as shape files of their geometry, are provided in Pangaea online data portal (Atanackov et al., 2022).

One of the important features of the new PSHA, compared to the old one (Lapajne et al., 2003) is that for the first time in Slovenia, a thorough parameterization of active faults was performed and a related database was established (Atanackov et al., 2021). The seismic activity of each fault source is defined by the slip rate parameter. Slip rates are usually estimated from all available data obtained with different methods covering very different time spans. Time spans range from several years (GNSS, PSInSAR, extensometer data), to several decades (leveling data), to thousands and tens of thousands of years (paleoseismic, geomorphic, geologic data) and even several million years (geologic data). Consequently, the slip rate data set is highly heterogeneous. As the tectonic moment rate is not released through earthquakes only, the slip realized through seismic activity has to be estimated. A fraction of slip in the frictional regime that occurs during earthquakes is named seismic coupling (i.e. Bird and Kagan, 2004, Carafa et al., 2017) or seismic efficiency (Basili et al., 2023). In this paper, the term seismic coupling is used. There are not many studies on determining seismic coupling to be used in PSHA, especially on active shallow crust tectonic regions. The papers that deal with seismic coupling in the region (Ward, 1998; Burrato et al., 2008; Bus et al., 2009; Carafa et al., 2017) were studied and discussed in Sec. 2.4.

This study aims to estimate seismic coupling and seismogenic depth layers, which significantly impact the activity rate of fault seismogenic sources but were only briefly described in Šket Motnikar et al. (2022). The PSHA procedure (e.g., Baker et al., 2021) requires the estimation of upper and lower seismogenic depths for all types of seismic sources (e.g., fault, area, and point sources), and in addition, the expected hypocentral depth for AS and PS should be provided. Also, the seismogenic thickness (difference between lower and upper seismogenic depth) is needed for estimating the seismic part of the slip rate (Bird and Kagan, 2004; Carafa et al., 2017). For the new Slovenian seismic hazard model, most of the parameters of the fault seismic source model are based only on the knowledge of active tectonic structures (Atanackov et al., 2021, 2022), since we kept the parameter estimation independent and uninfluenced by seismological data. On the other hand, the parameters for area source zones were mainly based on seismological data. Therefore, the seismogenic depth was determined using both geological and seismological approaches.

In the continuation, we first present in detail the relevant data and methods for geological and seismological estimation of seismogenic depth. Besides the available seismic coupling values from the literature, we adopted the method developed by Carafa et al. (2017). In the second part, we discuss the obtained results of upper and lower seismogenic depth and hypocentral





depth. Applying the obtained seismic coupling, we compared the total seismicity as estimated from seismological and geological data. The comparison was made in terms of annual activity rate and seismic moment rates, both for the entire observation area and individual area source zones.

## 2.    DATA AND METHODS

The first step in PSHA is a determination of the seismic source characterization (SSC) model that defines the spatial location

of future earthquakes, and their frequency and magnitude distribution. Seismic source characterization consists of developing a model that includes the historical earthquake catalogue, instrumental earthquake catalogue, and the development of a regional seismotectonic model. A key aspect of any SSC model building process is to consider all relevant and up-to-date seismotectonic, geological, and seismological data, models, and methods.

### 2.1.    Seismic source characterization models

The geological, geophysical, geotechnical, and seismological data are the basis for the development of the regional seismotectonic model (Fig. 1). Paleoseismic, geomorphic, and geological data are of special importance to PSHA because they provide information about the seismic activity associated with long recurrence intervals, that is not captured with earthquake

catalogues and geodetic data (Morell et al., 2020).

The active faults map and the corresponding database, compiled by Atanackov et al. (2021), cover the territory of Slovenia and include cross-border faults and faults in the near vicinity. Besides the earthquake catalogues, they represent the main data for the parametrisation of fault and area seismic sources (Fig. 2). The F source model includes all known active faults that are

able to generate an earthquake of $m_w$ 5.5 or higher. Each fault source is also given the probability of its activity in four categories (active 1.0, probable 0.7, potential 0.5, or questionable faults 0.25). The F source model consists of 89 fault seismic sources; 67 sources are parametrized by Atanackov et al. (2022), and the parameters for the remaining 22 fault sources were taken from the European Database of Seismogenic Faults (EDSF) (Basili et al., 2013).

Area source zones are the most standard type of seismic sources in PSHA. The A source model consists of 18 area source zones (Fig. 2) and covers the whole influential area (Šket Motnikar et al., 2022).



**Figure 2: Seismic source characterization models: fault seismic sources with the surface projection of fault plane (red colour), area seismic sources (black polygons) and point seismic sources (grey rectangular grid).**

## 2.2. Earthquake catalogues

The seismological information is based on two earthquake catalogues that were compiled for the new PSHA of Slovenia. The historical catalogue of larger earthquakes for Slovenia and the surrounding region (KPN2018) has a target completeness of $m_w$=3.5 (Šket Motnikar et al., 2022). There are 2867 earthquakes in the study area (12°E to 18°E and 44.5°N to 47.8°N). For the purpose of depth determination, the H_KPN18 catalogue was prepared, where only earthquakes from 1900 on were selected and earthquakes with unknown or zero depth were removed. There are 1477 earthquakes remaining for depth analysis in the H_KPN18 catalogue (black circles in Fig. 3).

**Figure 3: Earthquakes from H_KPN18 (black circles) and IR18 (yellow dots) catalogues used for depth determination of fault and area seismogenic sources.**

In general, reliable hypocentre information can only be derived from earthquakes recorded with modern instruments. For the

territory of Slovenia, we have used the earthquake catalogue of instrumentally recorded earthquakes in the period 2004 – 2018 (ARSO, 2018). To cover the whole study area of interest we have also used the seismic catalogues of neighbouring countries:

the Friuli catalogue (INOGS, 1977-2014) from 1980 on, the Austrian catalogue (ZAMG, 2002; 1998-2014) from 1990 on and BSHAP catalogue (BSHAP-2, 2015) from 1990 on. From catalogues for the neighbouring countries, we removed the earthquakes with unknown or zero depth. From the Slovenian catalogue, we removed the earthquakes with unknown depth or depth zero km, and poorly constrained earthquakes where the maximum azimuthal station gap is greater than 180 degrees and where the number of recording stations is smaller than 5. The 29018 earthquakes from all these catalogues were joined to the IR18 catalogue (yellow dots in Fig. 3).

The H_KPN18 and IR18 catalogues were used for seismogenic depth estimation, and consequently for seismic coupling estimation. The declustered catalogue KPN2018 was used for seismic moment rate calculation.

## 2.3. Seismogenic depth

Seismic source (e.g. fault, area, or point source) represents the source that could produce seismicity. Therefore the seismogenic part of the lithosphere has to be constrained in depth. The upper and lower limits are called upper and lower seismogenic depth and can be assessed either based on knowledge of active tectonic structures or by studying the depth distribution of past earthquakes. To take into account the uncertainty of earthquake depth determination, a chosen percentile depth cut-off was used. The seismogenic thickness (difference between lower and upper seismogenic depth) is also needed for the estimation of the seismic part of the slip rate.

The lower seismogenic depth was determined based on geological and seismological approaches. In Sec. 3, for each FS individually, we decided which among the two (geological vs. seismological) lower seismogenic depth estimates is used for assessing the seismic part of the slip rate. For AS, the comparison between seismological and geological lower seismogenic depth showed that seismological estimates are deeper at all AS except for one source zone. For each AS, the deeper estimate is chosen for the final lower seismogenic depth.

The upper seismogenic depth is estimated geologically for all FS and AS. Additionally, the expected hypocentral depth should be given for AS and PS. It was estimated from measures of central tendency using seismological data.

All depths in earthquake catalogues and all assessed depths of seismogenic sources refer to the WGS 84 reference ellipsoid.

### 2.3.1. Geological depth estimation

The upper seismogenic depth for FS is estimated from known geological (based on published data, fieldwork, and geomorphologic analyses) and geophysical data interpretation. We assume that all faults that exhibit clear surface traces can produce a surface rupture in the event of large earthquakes. For the buried faults we consider the depth to the subsurface fault


tip as visible in the geophysical profiles. For some faults outside Slovenia (Friuli), the upper depth is taken from the literature (Basili et al., 2013).


The upper seismogenic depth for AS is determined from geological information because there is no seismological information to support any other value. The upper seismogenic depth value for all AS is 0 km because the majority of fault sources exhibit surface traces and we assume that they can produce surface rupture in case of a large earthquake.

The upper seismogenic depth is therefore hypothetically set to 0 km both for FS and AS, except for known blind faults in the FS model. This is based on the assumption that the fault zones mapped at the surface were formed by surface rupturing earthquakes. However, exhumation processes bring on the surface also fault zones from previous tectonic phases that are currently inactive, so our assumption includes uncertainties, mainly related to the unknown relationship between the surface expression of faults and the underlying seismogenic structures. On the other hand, structurally immature faults may not have

easily recognizable fault traces at the surface, and in such a case, the FS has not been identified. The 0 km upper seismogenic depth can be considered confirmed for the faults that were recognized as active with one of the surface investigation methods (geomorphological offsets, outcrop observations, or shallow geophysics), testifying the surface rupturing paleoearthquakes.

The geological lower seismogenic depth estimation of the FS and AS is a comprehensive analysis incorporating various lines

of evidence, including published tectonic models, structural-geological and geophysical data, as well as the reinterpretation of these data sets in the light of derived seismic source models. One important factor considered in this estimation is the presence of detachments and their potential implications for seismic activity. Additionally, the intersection of the fault plane with other fault planes within the model and constraints on fault geometry inferred from seismicity distribution are taken into account. It is worth noting that the lower seismogenic depth of the nearest FS was adopted in the absence of any other data or argument.

This approach ensures a conservative estimate and serves as a reasonable starting point, considering the limited available information. However, it is important to recognize the inherent uncertainties associated with this approach and acknowledge the need for further investigations and data acquisition to refine the geological depth estimation in the future.

### 2.3.2. Seismological depth estimation

Seismological depth in the observed area was analyzed for various cell sizes and seismic source models ($5 \times 5$ km$^2$ cells, $7.5 \times 7.5$

km$^2$ cells, $10 \times 10$ km$^2$ cells, AS, FS buffers). For each of them and both catalogues (IR18 and H_KPN18) minimum, maximum, median, average, 90$^{th,}$ and 95$^{th}$ percentile of depth were calculated. To achieve robust depth estimates that are insensitive to outliers (occasional earthquake locations at large depths), a 95$^{th}$ percentile depth cut-off was applied.

The IR18 catalogue shows an uneven distribution of earthquakes either because of spatially heterogeneous earthquake activity

rates or because of different catalogue's origins (different threshold magnitude/intensity, seismic network density). The best





covered areas are Slovenia and Friuli, therefore, depth evaluation in these two areas is better than in other areas. We are aware of the problem that instrumentally recorded catalogues include mainly weak and moderate size earthquakes and that the large size earthquakes might not follow the same depth distribution. Therefore we also used the catalogue H_KPN18 with historical earthquakes to derive the second set of lower seismogenic depth estimates.


Based on this analysis, the seismological estimate of lower seismogenic depth in a given AS is chosen to be the deeper of the two 95th percentile (from IR18 and H_KPN18 catalogues) values. The exception is made where there is a large difference between the 95th and 90th percentile due to the small number of earthquakes. In such cases, the value of 95th percentile is strongly influenced by the few deepest earthquakes, for which the depth estimates are poorly constrained. In such cases, the

90th percentile of depth was chosen. If case of a single or no earthquakes inside the area source zone, the values for lower seismogenic depth from the neighbouring zone were adopted.

For seismological depth estimates of FS, we assessed the depth distribution of earthquakes with the IR18 catalogue only. For this purpose, the earthquake was attributed to the FS if it is less than 5 km from the surface projection of the FS plane (FS

buffer).

The values of the 50th percentile from IR18 and H_KPN18 don't differ much, but the H_KPN18 contains also strong earthquakes, therefore the expected hypocentral depth was determined as the 50th percentile (median) from the H_KPN18 catalogue. In case of a single or no earthquake inside the area source zone, the values for expected hypocentral depth from the

neighbouring zone were adopted.

### 2.4.  Slip rate and seismic coupling

The seismic activity of each fault source in the F model is characterized by two key components: the slip rate parameter and its associated seismic component. In this study, we have undertaken a comprehensive estimation of slip rates by utilizing a

diverse range of data obtained from various projects, each employing distinct methodologies and objectives. The data collection spans significantly different time scales, enabling a robust analysis. The time spans encompassed in our investigation vary from short-term observations of a few years, including data obtained from GNSS (Global Navigation Satellite System), PSInSAR (Persistent Scatterer Interferometric Synthetic Aperture Radar), and extensometer measurements, to medium-term observations spanning several decades, derived from levelling data. Additionally, we incorporated long-term datasets covering

timeframes of thousands and tens of thousands of years, acquired through paleoseismic, geomorphological, and geological investigations, data spanning several million years, sourced from geological records. Consequently, the slip rate data set is highly heterogeneous (Atanackov et al., 2021, 2022). For each individual FS, we provided the minimum, maximum, and best





estimate. It should be noted that the slip rates in the database of the Slovenian seismic source model (Atanackov et al., 2021, 2022) correspond to the total (seismic and aseismic) slip.


The seismic coupling $c$ is defined as the fraction of slip in the frictional regime that occurs in earthquakes (e.g. Bird and Kagan, 2004, Carafa et al., 2017). We used two approaches for the determination of seismic coupling. In the first approach, we reviewed the existing literature and adopted the best assessment for the area under consideration. In the second approach, we determined the seismic coupling individually for each FS following Carafa et al. (2017). The values thus obtained were

compared in terms of seismic moments.

The seismic moment of a given fault source is defined as the product of the shear modulus of the crust $\mu$, the area of the fault $A$, and the displacement $D$ (Aki, 1966):

$$M_{total} = \mu A D \, ,\tag{1}$$

The total moment rate $\dot{M}_{total}$ is the average seismic moment of the selected fault source in the given time period:

$$\dot{M}_{total} = \mu A S \, ,\tag{2}$$


where $S$ is the average slip rate.

Seismic moment $M$ released through an earthquake can also be determined from the earthquake size and is calculated from its moment magnitude $m_w$ using the Eq. (1) in (Kagan, 2002):


$$M = 10^{1.5(m_w+6)}.\tag{3}$$

For the seismological calculation of seismic moment rate (denoted as $\dot{M}_{seis}$) based on the earthquake catalogue, we followed Eq. (7) in Kagan (2002):


$$\dot{M}_{seis} = \frac{\alpha_0 m_t^\beta \Gamma(2-\beta)}{1-\beta} M_c^{1-\beta} exp\left(\frac{M_t}{M_c}\right) \, ,\tag{4}$$

where $M_t$ is the threshold moment for the completeness of the catalogue, $\alpha_0$ the total annual number of earthquakes above completeness magnitude $m_t$, $M_c$ the corner (maximum) moment and $\beta$ being the slope of the moment-frequency relation.





The tectonic moment rate (denoted as $\dot{M}_{tect}$), obtained from the geological data, was calculated with Eq. (5). Since not all of the fault energy is released through seismic events, the coupling parameter $c$ is added to the equation,

$$\dot{M}_{tect} = c\dot{M}_{total} = c\mu AS \; . \tag{5}$$

The F source model consists of 89 FS, among which 67 sources are parametrized by Atanackov et al. (2022), and the remaining 22 FS were taken from the European Database of Seismogenic Faults (EDSF) (Basili et al., 2013). The slip rate values in the EDSF database already correspond to the seismically coupled part and were used as such for ESHM20 (Danciu et al., 2021). Therefore, in the case of these 22 FS, in this study, we used the slip rate data as they are, without any correction of seismic coupling (c=1).

Several authors have extensively investigated seismic coupling in the region, including Ward (1998), Burrato et al. (2008), Bus et al. (2009), and Carafa et al. (2017). These studies employ different methodologies, resulting in diverse outcomes regarding seismic coupling. Table 1 provides a summary of select papers.

The variable estimates of seismic coupling can be attributed to several factors. Firstly, the presence of different tectonic regions within the study area may introduce variations in seismic coupling behaviour. Secondly, the seismic catalogue used in these 305 studies may suffer from incompleteness, particularly in capturing infrequent large events with long return periods. Additionally, limited observational periods of GNSS data collection may also contribute to uncertainties in estimating seismic coupling. Finally, the combination of all the aforementioned factors could collectively contribute to the observed variability in seismic coupling estimates.

**Table 1: The seismic coupling as interpreted from the most appropriate literature.**

| Author | Paper title | Considered area | Seismic coupling c (%) |
|---|---|---|---|
| Ward (1998) | On the consistency of earthquake moment rates and space geodetic strain rates: Europe | Italy<br>Balkan<br>world/Mediterranean | 71<br>59<br>50 |
| Bus et al. (2009) | Active crustal deformation in two seismogenic zones of the Pannonian region – GPS versus seismological observations | Central Pannonia<br>Mur – Murz zone | 17<br>34 - 58 |

### 2.4.1. Seismic coupling and coupling thickness of the seismogenic lithosphere

Carafa et al. (2017) proposed a method, which enables estimation of the seismic coupling individually for each fault seismic 315 source. The value depends on the coupled thickness of the seismogenic lithosphere ($cz$) which is defined as the product of the



difference between the upper and lower seismogenic depth (seismogenic thickness $z$) of the given fault source and its seismic coupling ($c$) for different fault kinematics (Table 2).

**Table 2: The coupled thickness of seismogenic lithosphere ($cz$) for different fault kinematics according to Carafa et al., 2017.**

|  | compressional faults | extensional faults | strike-slip faults |
|---|---|---|---|
| coupled thickness ($cz$) (km) | $3.7 \pm 0.7$ | $7.2^{+2.5}_{-1.5}$ | $4.8 \pm 0.9$ |


Considering the proposed method and values in Table 2, we estimated the seismic coupling c for each of the 67 FS.

For the calculation of the coupled thickness, we slightly corrected the estimation of the lower seismogenic depth for a few chosen FS (see Sec. 3). For the lower seismogenic depth, we used either the value from geological data (e.g. from the F model), applied for most FS, or the value determined from seismological data (e.g. from the A model). The criteria for choosing one or another value are described in the Sec. 3. The upper seismogenic depth was taken as estimated in the F model.

## 3. RESULTS AND DISCUSSION

### 3.1. Seismogenic depth

#### 3.1.1. Fault source model

The lower seismogenic depth was estimated based on geological and seismological data. To decide, which of the two values better determines the coupled thickness, we compared the geologically determined lower depth with the histogram that presents the spatial distribution of the earthquakes' depth from the IR18 catalogue in the FS buffer. Also, the 5[th] and 95[th] percentile of depth are shown (Fig. 4). We have evaluated each FS separately using expert knowledge and several criteria.

For most of the FS, we chose the geological depth estimates. The most typical argumentations and examples (Fig. 4) are explained as follows. If the earthquakes contained in the FS buffer are deeper than the geological depth estimate:

- these earthquakes were either believed to belong to other neighbouring FS that are overlapping with the assessing source (SS.SI-049 in Fig. 4), or
- the knowledge of active tectonic structures is poor and we are not able to attribute these earthquakes to any of FS (SS.SI-20 in Fig. 4), or
- the FS due to its limited length cannot accommodate deeper earthquakes in order to respect the fault geometric aspect ratio of the tectonic structure (SS.SI-035 in Fig. 4).

In some cases, the earthquakes originate on the margins of the tectonic structure and are not representative of this structure (SS.SI-40 in Fig. 4). If the seismicity in the FS buffer is not as deep as the geological depth estimate and the FS is prominent (regional) – we believe that the too short time span of seismological observations and mainly weak events in the IR18 catalogue





are not representative of the lower seismogenic depth of the structure (SS.SI.046 on Fig. 4). The geological depth estimates were also chosen for those FS where either geological depth estimates and earthquakes depth distribution match well (SS.SI-074a on Fig. 4) or earthquakes contained in the FS buffer are too few to reliably estimate the depth. The data published in EDSF (Basili et al., 2013) for the FS in the Friuli area are used as given in the database.

The seismologically determined lower seismogenic depth is a better choice for 12 FS because it can be assumed that the earthquakes are representing the behaviour of these structures better. This is true for the seismicity of prominent (also regional) structures where the earthquakes are considerably deeper than the geological depth estimates (SS.SI-019 in Fig. 4). Also the FS, for which the histograms show normal distribution, and the geological depth estimate is approximately at its median, we infer that the seismological data represent the behaviour of the structure better (SS.SI-032 on Fig. 4). The seismogenic lower
depth ranges from 5 km (SS.SI078 Northern Karavanke thrust fault) to 20 km (SS.SI-056a,b Sava W and E faults).

Geological and seismological values of upper and lower seismogenic depth for all FS in the F model are available in Pangaea online portal, columns *Min depth_geo, Max depth_geo, Max depth_seismo* (Atanackov et al., 2022).

**Figure 4: Some examples of IR18 catalogue earthquakes depth distribution within FS. Blue lines represent the 5th and 95th percentile respectively, and the red line denotes lower seismogenic depth based on knowledge of active tectonic structures.**




### 3.1.2. Area source model

The spatial distribution of the 95[th] percentile of depth from IR18 catalogue earthquakes in 5x5 km$^2$ cells (Fig. 5) shows a good correlation of earthquake epicentres with depth to Moho (e.g., Brückl et al., 2007; Grad et al., 2009; Stipčević et al., 2020). Deeper hypocentres in the northwest, west and central regions generally correspond to more profound Moho levels. However, some shallow earthquakes (Bajc et al., 2001; Zupančič et al., 2001) in the Julian Alps, the area of the deepest regional Moho, show discordance between the general hypocentre-Moho depth relation, pointing to the importance of dedicated local studies

and also a better knowledge of local active fault characteristics. The epicentres originating in the eastern part of Slovenia around Posavje, Krško Basin and Gorjanci Mountains are the shallowest in the areas where Moho decreases. Catalogue IR18 includes mainly weak and moderate size earthquakes, thus the depth distribution of large size earthquakes might show a different distribution. The spatial distribution of depth from historic earthquakes (H_KPN18 catalogue) in 5x5 km$^2$ cells is not representative due to a small number of earthquakes in individual cells.

The estimates of the lower seismogenic depth for AS using catalogue IR18 are shown in Table 3 and Fig. 5 and 6, while the results in Table 4 correspond to the catalogue H_KPN18. The final value of the lower seismogenic depth represents the deeper among the geological and seismological estimates. For all AS but one (AS8 Labot), the seismological estimate is taken. The final values of lower seismogenic depth are mainly influenced by the historical earthquakes from the H_KPN18 catalogue and only partially reflect the depth pattern described above. The values of upper and lower seismogenic depths, and of hypocentral

depths are given in Table 5 and are shown in Fig. 7.

**Table 3: Earthquake depth statistics per area source zone with IR18 catalogue (min, max, average, standard deviation, 50[th], 90[th], 95[th] percentile are in [km]).**

| Area source | No. of earthquakes | min | max | average | Standard deviation | 50[th] percentile (median) | 90[th] percentile | 95th percentile |
|---|---|---|---|---|---|---|---|---|
| AS1 | 8151 | 0.1 | 36.7 | 8.6 | 3.8 | 8.7 | 13.1 | 14.2 |
| AS2 | 6 | 7 | 15 | 11.2 | 2.9 | 11.5 | 14.5 | 14.8 |
| AS3 | 747 | 0 | 32.2 | 8.6 | 3.6 | 8.8 | 12.1 | 13.3 |
| AS4 | 185 | 0.3 | 34 | 11 | 5.5 | 9.9 | 18.3 | 22.1 |
| AS5 | 10438 | 0 | 25.2 | 9.2 | 4.5 | 8.9 | 15.2 | 15.8 |
| AS61 | 1930 | 0 | 18.7 | 6.9 | 2.9 | 7.1 | 10.4 | 11.2 |
| AS62 | 1585 | 0 | 23.4 | 5.0 | 3.2 | 4.9 | 8.5 | 10.5 |
| AS7 | 23 | 5 | 22 | 9.8 | 4.2 | 9.0 | 14.8 | 15.9 |
| AS8 | 81 | 0.1 | 10.9 | 6.7 | 2.3 | 6.9 | 9.2 | 9.7 |
| AS9 | 98 | 0.2 | 40 | 10.1 | 5.9 | 10 | 16.4 | 17.4 |
| AS10 | 34 | 0 | 18.1 | 7.5 | 4.4 | 9.0 | 11.1 | 12.9 |
| B1 | 134 | 1.1 | 51.1 | 17.3 | 7.8 | 16.6 | 26.1 | 30.7 |
| B2 | 237 | 0.1 | 35.2 | 8.2 | 5.2 | 7.8 | 13.9 | 16.5 |



| B31 | 1 | | 12.2 | 12.2 | 12.2 | 0 | 12.2 | 12.2 | 12.2 |
| B32 | 9 | | 5.9 | 13 | 9.3 | 1.9 | 9.0 | 11.4 | 12.2 |
| B4 | 29 | | 0.7 | 32 | 9.0 | 6.2 | 8.0 | 11.4 | 20.8 |
| B5 | 5 | | 0 | 20.3 | 9.9 | 7.9 | 8.0 | 18.2 | 19.2 |
| B6 | 0 | | | | | | | | |

**Table 4: Earthquake depth statistics per area source zone with H_KPN18 catalogue (min, max, average, standard deviation, 50th, 90th, 95th percentile are in [km]).**

| Area source | No. of earthquakes | min | max | average | Standard deviation | 50th percentile (median) | 90th percentile | 95th percentile |
|---|---|---|---|---|---|---|---|---|
| AS1 | 257 | 0.4 | 51 | 9 | 5.4 | 9 | 13.4 | 15 |
| AS2 | 10 | 7 | 18 | 9.7 | 3.3 | 8 | 14.4 | 16.2 |
| AS3 | 55 | 1.5 | 22 | 9.8 | 4.4 | 8.6 | 15.2 | 17.6 |
| AS4 | 97 | 0.3 | 33 | 10.3 | 5.4 | 10 | 17 | 18.4 |
| AS5 | 222 | 0.5 | 99 | 9.8 | 9.4 | 8.1 | 15.7 | 19 |
| AS61 | 58 | 1 | 18.3 | 7.6 | 4 | 7 | 13.2 | 16.1 |
| AS62 | 186 | 0.1 | 26 | 8.5 | 5.7 | 6.9 | 18.2 | 19.9 |
| AS7 | 77 | 4 | 22 | 7.8 | 2.5 | 8 | 10 | 11 |
| AS8 | 28 | 4.4 | 15.6 | 8.7 | 2.2 | 8 | 12 | 12.1 |
| AS9 | 140 | 0.1 | 30 | 10.8 | 6.2 | 10 | 18.1 | 21.1 |
| AS10 | 31 | 3.3 | 29.6 | 12 | 7 | 11.6 | 19.4 | 25.3 |
| B1 | 11 | 1 | 18.3 | 11 | 5.9 | 10 | 18.3 | 18.3 |
| B2 | 8 | 0.9 | 11 | 6.2 | 3.6 | 5 | 10.5 | 10.8 |
| B31 | 0 | | | | | | | |
| B32 | 19 | 3 | 16 | 8.1 | 3 | 8 | 11.4 | 13.3 |
| B4 | 60 | 1 | 49 | 8 | 5.9 | 8 | 10 | 11 |
| B5 | 35 | 0.1 | 17 | 8.4 | 3.1 | 8 | 13.1 | 14.1 |
| B6 | 1 | 7 | 7 | 7 | 0 | 7 | 7 | 7 |
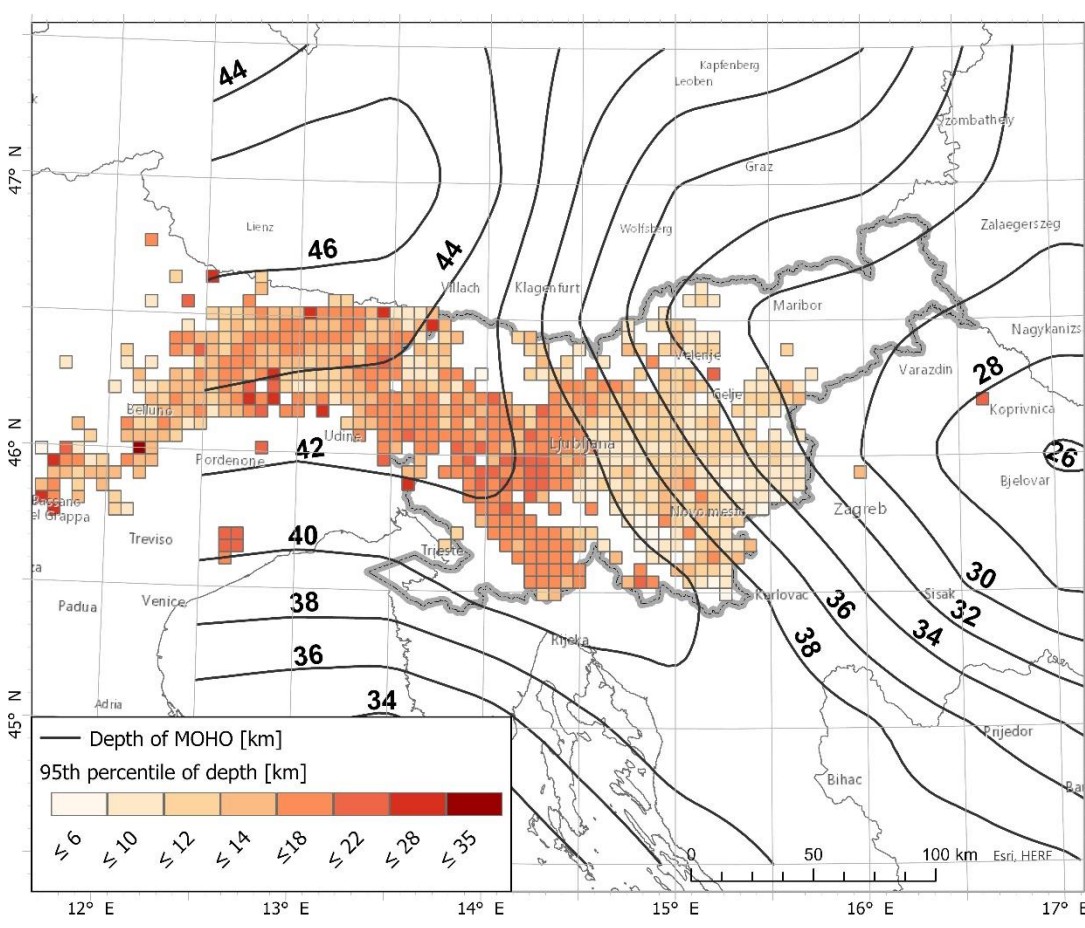

**Figure 5: The spatial distribution of 95th percentile of depth from IR18 catalogue earthquakes in 5x5 km² cells. Only cells containing**
**at least four earthquakes are shown. Contours represent the depth of Mohorovičić discontinuity (Brückl et al., 2007; Grad et al.,**
**2009).**





**Figure 6: Earthquakes depth distribution per area source in IR18 catalogue (blue lines represent 5th and 95th percentile in [km] respectively), and the red dashed line represents the geological lower depth estimate. The B6 South Hungary area source contains no earthquakes. The scale for the number of earthquakes is logarithmic.**





**Table 5: Assigned final values of upper and lower seismogenic depth and hypocentral depth for A model.**

| Area source | Area source name | Lower depth geological [km] | Lower depth seismological [km] | Final lower depth [km] | Hypocentral depth [km] | Upper depth [km] |
|---|---|---|---|---|---|---|
| AS1 | Friuli | 10 | 15 | 15 | 9 | 0 |
| AS2 | Molltal | 15 | 16 | 16 | 8 | 0 |
| AS3 | Periadriatic | 15 | 18 | 18 | 9 | 0 |
| AS4 | Outer Dinarides | 12 | 22 | 22 | 10 | 0 |
| AS5 | Dinarides | 15 | 19 | 19 | 8 | 0 |
| AS61 | Posavje | 12 | 16 | 16 | 7 | 0 |
| AS62 | Gorjanci | 10 | 19 | 19 | 7 | 0 |
| AS7 | Mur-Murz-VBTF | 15 | 16 | 16 | 8 | 0 |
| AS8 | Labot | 15 | 12 | 15 | 8 | 0 |
| AS9 | Inner Dinarides | 15 | 21 | 21 | 10 | 0 |
| AS10 | Mid-Hungarian | 10 | 19 | 19 | 12 | 0 |
| B1 | North Adriatic | 12 | 31 | 31 | 10 | 0 |
| B2 | North Friuli | 15 | 17 | 17 | 5 | 0 |
| B31 | Carinthia 1 | 12 | 16 | 16 | 8 | 0 |
| B32 | Carinthia 2 | 12 | 13 | 13 | 8 | 0 |
| B4 | North Alps | 15 | 21 | 21 | 8 | 0 |
| B5 | Styria | 12 | 19 | 19 | 7 | 0 |
| B6 | South Hungary | 15 | 19 | 19 | 12 | 0 |


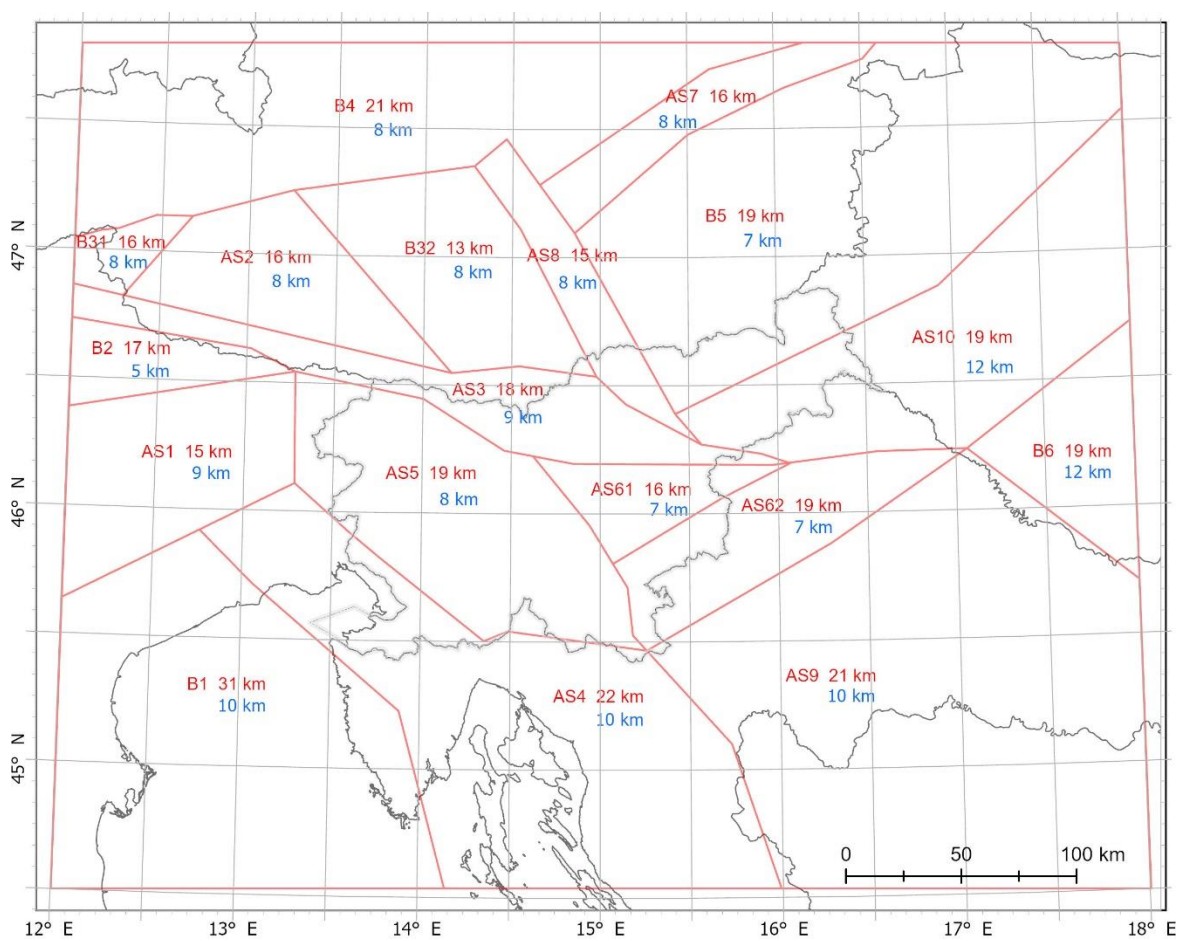

**Figure 7: Area sources lower seismogenic depth (red colour text) and hypocentral depth (blue colour text).**

### 3.2. Seismic moment rate calculation

We calculated the seismic moment rate $\dot{M}_{seis}$ from different complete earthquake sub-catalogues of KPN2018 using Eq. (4). We have varied the threshold magnitude $m_t$ 3.8 and 4.1, according to the completeness year 1875, $m_t$ 4.5 for the complete sub-catalogue since 1850, and $m_t$ 5.1 for the complete sub-catalogue since 1840. Such threshold magnitudes were chosen because moment magnitude $m_w$ strongly depends on the conversion from the intensities ($m_w$= 3.8 corresponds to intensity V EMS-98, 4.1 to intensity V-VI, 4.5 to intensity VI, 5.1 to intensity VII, and 5.6 to intensity VIII). The total annual number $\alpha_0$ of

earthquakes above $m_t$ for complete sub-catalogues is given in Table 6. For each complete sub-catalogue, we investigated the parameter space of $\beta$ and $M_c$ for a tapered Gutenberg-Richter distribution and selected the corner magnitude $m_c$ where the peak of the likelihood function is found (see Jackson & Kagan, 1999; Bird & Kagan, 2004 for the formal statement of the likelihood function and applications at the global scale).



**Table 6: Corner magnitudes for four complete sub-catalogues.**

| $m_t$ | Year of completeness | $\alpha_0$ | $m_c$ |
|---|---|---|---|
| 3.8 | 1875 | 6.92 | 6.52 |
| 4.1 | 1875 | 3.19 | 6.22 |
| 4.5 | 1850 | 1.82 | 6.72 |
| 5.1 | 1840 | 0.43 | 6.62 |

In the first sensitivity analysis (Table 7(a)), the corner magnitude is fixed to 6.7, which is the best estimate of maximum magnitude for the largest part of Slovenian territory in A model, and is also the best estimate of maximum magnitude in P model (Šket Motnikar et al., 2022). The corresponding corner moment $M_c$ was calculated using Eq. (3). We varied the

parameter $\beta$ approximately in the range, determined by Bird and Kagan (2004) for continental convergent boundaries, and Carafa et al. (2017) for compressional and strike-slip faults. The value of the shear modulus is 35.2 GPa (Carafa et al, 2017). Thus we calculate the $\dot{M}_{seis}$ [1E+17 Nm/year] for various $\beta$ and $m_t$ (Table 7a and Fig. 8(a)).

In the second sensitivity analysis (Table 7(b)), besides the varying parameter $\beta$, we also varied the parameter $m_c$ as obtained from Table 6.


**Table 7. The seismic moment rate $\dot{M}_{seis}$[1E+17 Nm/year] for various $\beta$ and $m_t$. Table 7(a): fixed $m_c$ as $m_{max}$ (6.7); Table 7(b): $m_c$ varies according to different $m_t$ as computed (Table 6).**

(a)

| Year of compl. | $\beta$ $m_t$ | 0.55 | 0.57 | 0.59 | 0.61 | 0.63 | 0.65 | 0.67 | 0.69 | 0.71 | 0.77 | 0.83 |
|---|---|---|---|---|---|---|---|---|---|---|---|---|
| 1875 | 3.8 | 6.19 | 5.30 | 4.56 | 3.93 | 3.39 | 2.94 | 2.56 | 2.24 | 1.96 | 1.37 | 1.04 |
| | 4.1 | 5.05 | 4.41 | 3.87 | 3.41 | 3.00 | 2.66 | 2.36 | 2.11 | 1.89 | 1.41 | 1.13 |
| 1850 | 4.5 | 6.16 | 5.54 | 4.99 | 4.51 | 4.09 | 3.73 | 3.40 | 3.12 | 2.88 | 2.33 | 2.03 |
| 1840 | 5.1 | 4.56 | 4.28 | 4.02 | 3.79 | 3.58 | 3.40 | 3.23 | 3.09 | 2.97 | 2.72 | 2.69 |

(b)

| Year of compl. | $\beta$ $m_c$ | 0.55 | 0.57 | 0.59 | 0.61 | 0.63 | 0.65 | 0.67 | 0.69 | 0.71 | 0.77 | 0.83 |
|---|---|---|---|---|---|---|---|---|---|---|---|---|
| 1875 | 6.52 | 4.68 | 4.06 | 3.53 | 3.08 | 2.70 | 2.37 | 2.08 | 1.84 | 1.64 | 1.19 | 0.93 |
| | 6.22 | 2.39 | 2.17 | 1.96 | 1.78 | 1.63 | 1.49 | 1.37 | 1.26 | 1.17 | 0.96 | 0.85 |





| 1850 | 6.72 | 6.35 | 5.70 | 5.14 | 4.64 | 4.20 | 3.82 | 3.48 | 3.19 | 2.93 | 2.37 | 2.06 |
| 1840 | 6.62 | 4.03 | 3.80 | 3.59 | 3.41 | 3.24 | 3.09 | 2.96 | 2.84 | 2.74 | 2.56 | 2.57 |

### 3.2.1. Comparison of seismic and tectonic moment rates for all FS

Besides the calculations of the seismic moment rate $\dot{M}_{seis}$ (based on seismicity and with various $\beta$, $m_{c,}$ and $m_t$), we calculated also the tectonic moment rate $\dot{M}_{tect}$. For seismogenic fault sources from EDSF (Basili et al., 2013), the seismic coupling c is not applied. For the remaining 67 FS, we estimated five alternatives: fixed value c = 1, a fixed value of 0.7, and three (upper, mean, and lower) individually determined c values, according to Table 2 (as described in Sec. 2). Using five alternatives of
seismic coupling and applying Eq. (5), we calculated five alternatives of $\dot{M}_{tect}$ for all FS, and summed them up. The overall tectonic moment rates for all five alternatives are shown in Table 8 and presented with a red line in Fig. 8a and 8b, corresponding to the seismologically determined seismic moment rate from Table 7a and 7b respectively.

**Table 8. The sum of tectonic moment rates of all FS $\dot{M}_{tect}$ [in 1E+17 Nm/years] for five alternative approaches of seismic coupling.**
**Seismic coupling was not applied to FS taken from EDSF (Basili et al., 2013).**

| Seismic coupling c = 1 | Seismic coupling c = 0.7 | Individual coupling (Table 2, max estimate) | Individual coupling (Table 2, mean estimate) | Individual coupling (Table 2, min estimate) |
|---|---|---|---|---|
| 8.29 | 6.31 | 4.15 | 3.76 | 3.37 |


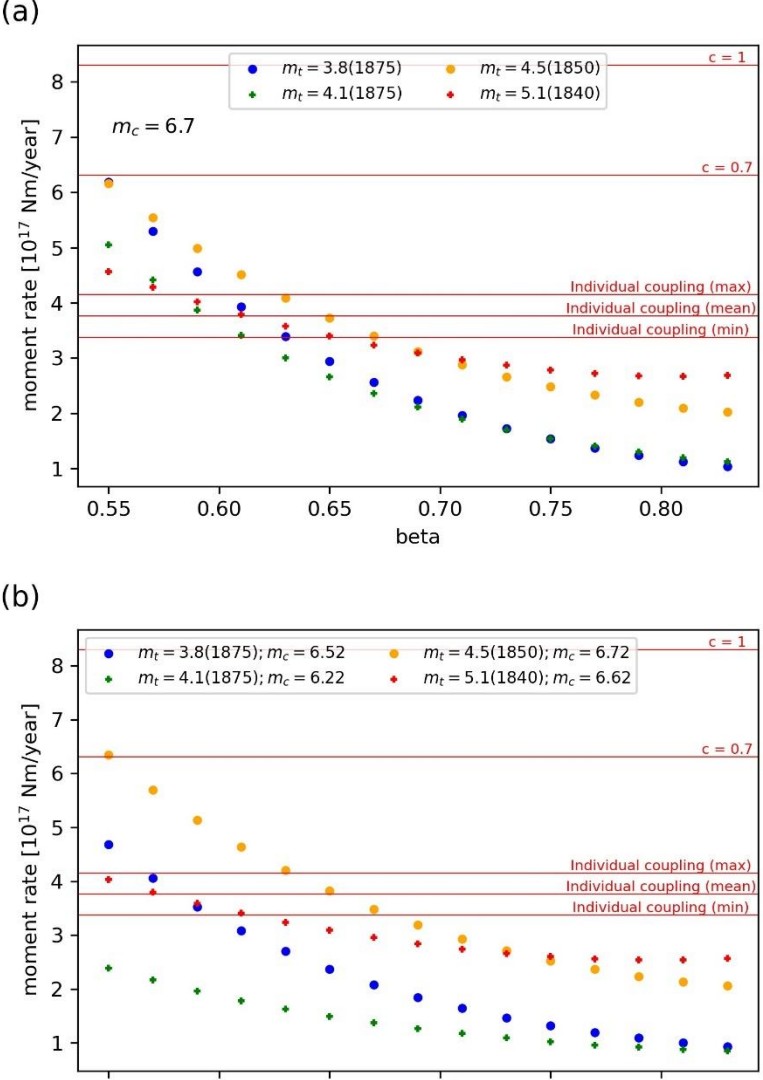

**Figure 8: Comparison of seismologically determined seismic moment rates $\dot{M}_{seis}$ (dots) and tectonic moment rates $\dot{M}_{tect}$ (red lines) for different threshold magnitude $m_t$ and different year of completeness from catalogue KPN2018 depending on the slope of the moment-frequency relation value β. Figure 8a: fixed $m_c$ = 6.7; Figure 8b: $m_c$ varied (values in Table 6).**

We compared seismologically determined seismic moment rates $\dot{M}_{seis}$ for different threshold magnitudes $m_t$ and different year of completeness of catalogues with tectonic moment rates. The results (Tables 7 and 8, and Fig. 8) show that the application of coupling is justified and the choice of method (after Carafa et al., 2017) is appropriate. The fit is the best between the seismic moment rates for $m_t$ 4.5 (yellow dots in Fig. 8) and for seismic couplings (mean estimate) after Carafa et al., 2017. Therefore,





a higher weight (70 %) is given to the "Carafa c" branch in PSHA, complementary to the branch that corresponds to the fixed c = 0.7 (weight 30 %).

### 3.2.2. Comparison of seismic and tectonic moment rates for FS, grouped by area source zones

A comparison of seismic moment rates calculated from the catalogue (threshold magnitude $m_t$ 4.5, year of completeness 1850, corner magnitude $m_c$ 6.7) vs. tectonic moment rates calculated from fault slip-rates is done for earthquakes and faults grouped by AS (Fig. 2).

Some representative results are shown in Fig. 9. For Posavje and Gorjanci areas the moment rates from seismicity since 1850
are way over the moment rates calculated from fault slip rates. This might suggest some hidden active structures that were not recognised and included in our fault-based SSC model. On the other hand, the tectonic moment rate is higher than the moment rate from seismicity in the Periadriatic zone. This could indicate very aseismic movement in the area. The moment rates from Dinarides and Outer Dinarides area zones show good agreement with the chosen coupling values. All FS in the Friuli zone are from EDSF (Basili et al., 2013), which is also reflected in graphs (c = 1). Results are a good indicator of areas, where future
investigations of active structures are necessary. They also indicate areas, where future analysis and estimation of slip rates and seismic coupling should be performed more in detail.






**Figure 9: Comparison between seismologically determined seismic moment rate $\dot{M}_{seis}$ (yellow dots) of individual area source zones and tectonic moment rate $\dot{M}_{tect}$ (red lines) for FS, grouped by area source zones. Seismic moment rate from the complete KPN2018 catalogue, $m_t$ 4.5, year of completeness 1850, and corner magnitude $m_c$ 6.7.**





### 3.3. Slip rate and seismic coupling

The seismic coupling in Carafa et al. (2017) was determined for faults grouped according to their kinematics. In that case, the authors didn't use the fault slip rates as direct input because they were often missing or with overlooked uncertainty (Carafa et al., 2022). Instead, after anticipating and discounting short-term transients (Carafa and Bird, 2016), the GNSS-derived horizontal strain rate tensor was a good proxy for them and was used in seismic coupling estimation by Carafa et al. (2017). In general, GNSS derived velocity field represents the bulk displacements; seismic slip and aseismic slip, off-fault

accumulation, and non-tectonic displacement. In our case, slip rate estimates refer to the total slip, including the aseismic part (Atanackov et al., 2021) and GNSS derived data represent only a minor part of input data for slip rate estimation.

To check the assumption that our estimation of the total slip rate and GNSS measurements do not differ much, we considered the largest parallel faults between (and including) Črni Kal – Palmanova and Idrija FS (area indicated with a dashed rectangle in Fig. 11) that is a proxy of the displacement in N-S direction (1,94 mm/year) which is the approximate regional stress

direction. Five of the seven chosen FS have the probability of activity 1. The other two fault sources have a probability of 0.7 (Buzet) and 0.5 (Divača). Considering Buzet and Divača as not active, which presents the lower limit of our estimate, the budget of slip in the N-S direction is reduced to 1,79 mm/year. The area was chosen also because in the western part of Slovenia, the GNSS network is the densest and therefore the observations are the most reliable. The most up-to-date review of measured GNSS vectors is available in the paper by Serpelloni et al. (2016). Interpolated velocity field of northward

movements between Črni Kal – Palmanova and Idrija Faults show a decrease of velocity towards the North from approximately 2.5 mm/year in the Slovenian Istra to 1.0-1.5 mm/year at the southern margin of the Julian Alps. Based on the GNSS data the total N-S horizontal shortening absorbed by the faults in this area at present is therefore approximately 1.0-1.5 mm/year. Since the interpolation is done on a relatively small number of GNSS data points, the estimated shortening is relatively poorly constrained and may be over- or under- estimated. The N-S shortening across the whole of Slovenia is estimated at 2-4 mm/year

according to the GNSS data from an earlier study (Weber et al., 2010).

The calculated N-S components of the estimated slip of the chosen faults are in Table 9. The comparison with the results of Serpelloni et al. (2016) and Weber et al. (2010) shows that the regional geologic and geodetic shortening rates are comparable.

**Table 9: Sum of N-S components of fault movements along the largest faults in western Slovenia.**

| Name | Fault ID | Dip | Rake | Strike | Slip [mm/year] | N-S projection of SR [mm/year] |
|------|----------|-----|------|--------|----------------|--------------------------------|
| Buzet | SS.SI-002 | 20 | 90 | 310 | 0.050 | 0.036 |
| Črni Kal - Palmanova | SS.SI-006 | 25 | 90 | 315 | 0.200 | 0.128 |
| Divača | SS.SI-011 | 80 | 160 | 305 | 0.200 | 0.118 |
| Raša | SS.SI-014 | 85 | 170 | 315 | 0.700 | 0.495 |
| Predjama-Avče | SS.SI-019 | 80 | 170 | 310 | 0.700 | 0.459 |
| Idrija | SS.SI-022 | 85 | 165 | 310 | 1.000 | 0.638 |





| Ravne | SS.SI-025 | 80 | 170 | 310 | 0.100 | 0.066 |
|---|---|---|---|---|---|---|
| | | | | | **Sum [mm/year]:** | 1.940 |


The seismic coupling value calculated following Carafa et al. (2017) depends on the seismogenic thickness of the given fault source and its fault kinematics. The mean estimate of seismic coupling ranges from 0.8 for Divača (SS.SI-011) shallow strike-slip fault to 0.24 for deep Periadriatic E and W faults (SS.SI-056a, b). The uncertainty in the seismic coupling of slip rates is

handled with two logic-tree branches. The first branch is based on the literature data (Table 1) and considers fixed 0.7 of the slip rates to be seismically coupled. The second branch is based on individual values of seismic coupling for each fault, calculated after Carafa et al. (2017). The weight of the first branch is 30 % and the weight of the second branch is 70 % (Šket Motnikar et al., 2022). The seismic coupling was applied to all but 22 FS from EDSF (Basili et al., 2013). The final estimate of seismic coupling ranges from 0.77 to 0.38 (Fig. 10).


The uncertainty in slip rate is modelled in three logic tree branches for lower, best, and upper alternative values. The seismic part of slip rate (best) estimates ranges from 0.01 mm/year to 0.65 mm/year. (Fig. 11). The highest seismic part of slip rates in Slovenia is estimated for the Sava (E and W) and Idrija faults with 0.43 mm/year and 0.42 mm/year respectively. The results of the slip rate estimates and seismic coupling for all FS in the F model are available in the Pangaea portal, columns *Min slip*

*rate, Max slip rate, Best estimate slip rate, Seismogenic layer thickness, Coupled thickness_Carafa mean, Coupled thickness_Carafa min, Coupled thickness_Carafa max, Seismic coupling_Carafa mean, Seismic coupling_Carafa min, Seismic coupling_Carafa max, Seismic coupling* (Atanackov et al., 2022).






**Figure 10: Individual seismic coupling factor for each fault seismic source.**







**Figure 11: The seismic part of the slip rates on individual faults as considered in the final PSHA calculation. The dashed rectangle indicates the area of slip rate analysis (Table 9).**

### 3.4. Comparison of total activity rate in the A, F, and P model

The three global seismic source models (A, F and P) have different purposes and assumptions, but each of the models should be complete by itself. It is expected that they produce different seismic hazard values and different areas of the highest values, although we cannot tell in advance which will give the highest hazard values and where. However, if one model significantly differs from others, it is recommended to analyze the most influential parameters and find reasons for such a difference. In Table 10, the total activity rate (annual number of all earthquakes above magnitude 0 in the influential area) of the three global

models is compared. In addition, four branches of the fault model (active, probable, potential, and questionable) are considered separately, and their weights are taken into account in the total values.





As expected, the difference between the A and P models is very small because both total activity rates are based on counting earthquakes in the complete (declustered) catalogue (from 1875, $m_t$=3.8). The difference of 0.5 % results from smoothing around borders of the influence area, and on different $m_{max}$ values in A and P models.

The F model gives a different picture of seismicity than the catalogue because the period of the complete catalogue is shorter than the return period of the largest possible magnitude of fault sources. Activity rate in the F model is converted from slip rate using Youngs and Coppersmith (1985) relationship, or (equivalently) the fourth model in (Bungum, 2007), Eq. (10), where the activity rate is explicitly expressed as:

$$N_4(m^0) = \frac{\mu A_f S(d-b)\left[1-e^{-\beta(m^u-m^0)}\right]}{bM^u e^{-\beta(m^u-m^0)}},\tag{6}$$

where $A_f = LW$ is fault area depending on depth and dip, $S$ is slip rate, $m^u$ is maximum magnitude, $m^0$ is lower bound magnitude, and $M^u$, $d$, $b$, and $\mu$ are constants. Details and values of constants are provided in Youngs & Coppersmith (1985) (Eq. (11)). When the slip rates are converted to activity rate, the total activity rate of the F model is approximately 62 % higher than the A model. We know that this is unrealistic because the slip rates estimated by Atanackov et al (2021, 2022) contain also the aseismic part and large events post-seismic parts. When the seismic coupling is 0.7, the total activity rate of the F model is approximately 24 % higher than the other two models A and P. When the seismic coupling is calculated after Carafa et al., (2017) individually per each fault source, the total activity rate of the F model is around 23 % lower than the other two models.

In the hazard calculation, we used two branches of seismic coupling with weights 70 %, and 30 %, respectively for coupling c = 0.7, and for coupling calculated after Carafa et al. (2017). Taking into account the two weighted branches, the total activity rate of the F model (grey line in Table 10) is around 9 % lower than the other two models A and P. This could be partly explained by the fact that some seismicity is generated on smaller faults that are not part of our F model. Also, the fault source buffers do not cover the whole area of observation. Therefore, in the hazard calculation, the F model is complemented with the background seismicity.

Despite the very different approaches of estimation and very different time spans of geological vs. seismological data, the seismic activity of all three global seismic models shows less than a 10 % difference, which does not indicate the need to revise the models.

**Table 10: Comparison of the total activity rate (annual number of earthquakes above magnitude 0 calculated from complete catalogue $m_w$>=3.8, 1875 on) in seismic source models. The middle column denotes weight according to the type of fault source activity (active, probably, potentially, or questionably active). In the F model, two alternatives of seismic coupling (c = 0.7, and mean seismic coupling after Carafa et al. (2017) are considered (grey line).**



| Model | Weighted N(0) of earthquakes | Weight according to the fault activity | Total N(0) of earthquakes |
|---|---|---|---|
| **A** | 39017 | | |
| **P** | 39234 | | |
| **F total (seismic coupling c = 1)** | 63468 | | 78307 |
| F active only | 42496 | 1 | 42496 |
| F probable only | 12106 | 0.7 | 17294 |
| F potential only | 8473 | 0.5 | 16947 |
| F questionable only | 393 | 0.25 | 1571 |
| **F total (seismic coupling c = 0.7)** | 48544 | | 60770 |
| F active only | 31557 | 1 | 31557 |
| F probable only | 9292 | 0.7 | 13275 |
| F potential only | 7419 | 0.5 | 14838 |
| F questionable only | 275 | 0.25 | 1100 |
| **F total (mean seismic coupling after Carafa)** | 30002 | | 39298 |
| F active only | 17883 | 1 | 17883 |
| F probable only | 5534 | 0.7 | 7906 |
| F potential only | 6414 | 0.5 | 12828 |
| F questionable only | 170 | 0.25 | 681 |
| **F total as calculated in the hazard model (70 % for 0.7 seismic coupling, and 30 % for seismic coupling after Carafa et al. (2017)** | 35565 | | |

## 4. CONCLUSION

The characterization of seismogenic sources and the process of determining the seismic coupling require knowledge of the seismogenic lithosphere's thickness, which is bounded by upper and lower seismogenic depth. This study, based on two updated earthquake catalogues for the transition zone between the Alps, Dinarides and Pannonian Basin, explores the
variability of seismogenic depth and seismic coupling to be included in the new Slovenian seismic hazard model. The detailed analysis of seismogenic depth is based on geological expert knowledge and on a seismological approach by studying the depth distribution of earthquakes for each FS and AS separately. The seismogenic lower depth for FS is in the range of 5 km (SS.SI078 Northern Karavanke thrust fault) to 20 km (SS.SI-056a,b Sava W and E faults). The estimated values of seismogenic lower depth for AS are higher and reflect the use of the historical H_KPN18 catalogue. The values for AS are in the range of
13 km to 31 km. The upper seismogenic depth for FS and AS is estimated from known geological and geophysical data interpretations. The upper seismogenic depth value for FS is in the range from 0 km to 6 km while it is 0 km for all AS. Seismogenic depth determined from the analysis of the earthquake catalogues depends greatly on the velocity model used to locate earthquakes. As Rajh et al. (2022) and Rajh (2022) demonstrated, an improved velocity model in our study area can have a profound effect on earthquake locations, especially their depths. The earthquake hypocentres relocated with the new

models were on average shallower by about 2-3 km. This could close the gap between the geological and seismological lower seismogenic depth estimates even further. However, at the time of this study, the models were not readily available.

Seismic coupling determines the proportion of slip rate that directly affects the annual earthquake rate in a given fault source and is therefore of great importance to PSHA. We used two approaches for the determination of seismic coupling. We reviewed

the existing literature and adopted the best assessment for the studied area and the chosen value of 0.7 was used as one logic tree branch in PSHA. In the second branch, the seismic coupling was estimated individually for each FS following Carafa et al. (2017). The values of seismic coupling following Carafa et al. (2017) depend upon the seismogenic layer thickness of the given fault source and the fault kinematics. The estimated seismogenic thickness for FS ranges from 5 km to 20 km and the weighted average of both estimates of seismic coupling ranges from 0.77 to 0.38. The values thus obtained were compared in

terms of seismic moment rates for the whole studied area and for individual FS grouped by AS. The sum of tectonic moment rates of all FS for alternative approaches of seismic coupling ranges from 3.76E+17 Nm/year for the mean estimate of individual coupling to 8.29E+17 Nm/year for c=1. We also compared the total activity rate based on the complete earthquake catalogue with the rate based on the seismic part of slip rates in FS and seismic activity of all three global seismic models shows less than 10 % difference. These comparisons complete our analyses of the seismic coupling estimation to national-

scale high-quality datasets.

*Data availability*

The data used in this manuscript are published as a dataset identified by the DOI https://doi.org/10.1594/PANGAEA.940100, which is openly accessible.


*Author contributions*

All authors contributed to the study's conception and design. The first draft of the manuscript was written by PZ and BŠM, and all authors read, commented, and improved previous versions of the manuscript. All authors approved the final manuscript.

*Competing interests*

The authors declare that they have no conflict of interest.

*Acknowledgments*





This research has been partly supported by the Slovenian Research Agency under Research Programs No. P1-0011 and P1-0419 and Young Researcher grant no. 1000-21-0510 .

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
