# Peer review of "Seismogenic depth and seismic coupling estimation in the transition zone between Alps, Dinarides and Pannonian Basin for the new Slovenian seismic hazard model"

_Natural Hazards and Earth System Sciences, 2023_

## Author Response (AR2)

**Response to reviews**

**Response to Editor's final decision:**
Thank you for a very thorough review and useful suggestions that made our paper better. Regarding the remark on figure adjustments: where the resolution was low we made new figures (in pdf) and included them in zip file.

**Response to Referee #1 comment:**

Thank you for reviewing our manuscript. The list of abbreviations will indeed help the reader therefore it is added to the end of the manuscript.

**Response to Referee #2 comment:**

GENERAL COMMENT
The paper is very good. It addresses an important aspect, not yet researched, related to seismic sources in Slovenia. The width and depth of the seismogenic layer are a crucial input in modern PSHA. It is well written and deserves publication. There are only a few details that should be improved in order to fully understand some parts, also for the non-specialized reader. I suggest therefore a minor review.
Thank you for recognising the importance of our work and for a very thorough review and many helpful suggestions and comments, that helped to improve the quality and readability of the manuscript. We have tried to cover all of your comments and suggestions and changed the manuscript accordingly. Replies to the comments are marked in red.

SPECIFIC COMMENTS (L = line of paper)
L18 Should add an introductory statement stating that three source models are used.
Added sentence: "The hazard model was based on three seismic source models: area source model, fault source model and smoothed seismicity (point) source model."

L74 We applied...No application of PSHA in this paper. Should say:Šket Motnikar et al. (2022) applied ...
Changed.

L77-78 Sentence has no clear meaning. Rewrite.
Changed. "The development of the Slovenian hazard model ran parallel to the update of the European seismic hazard model (ESHM20) project (Danciu et al., 2021)."

L104 The abbreviation PS is not explained before (e.g. in lines 81-85)
Added sentence in L87. "Point sources (PS) are centres of grid cells with 10 x 10 km dimensions, which cover the whole influence area of the hazard calculation (Šket Motnikar et al., 2022)."

L145 In the caption of Fig.2 there should be a reference (e.g. DOI Pangaea dataset) where is it possible to correlate the acronym of the fault with the actual fault name and properties.
Added sentence to Fig2 caption." The list of fault source acronyms, names and basic tectonic characteristics are given in the supplement Table S1, the full parametrisation for all fault and area seismic sources is available at https://doi.pangaea.de/10.1594/PANGAEA.940100."
We also added the fault and area source names besides acronyms to histograms in Fig4 and Fig6.

L162 If the ZAMG catalog spans the period 1998-2014 how is possible to consider events from 1990 on ?
The period from 1990 to 1998 is covered by catalogue ZAMG, 2002.

L236 »... from neighbouring zone...« Missing criterium of selection if there are several neighbouring zones !
Sentence rewritten. ".. from the neighbouring zone with the most similar tectonic characteristics were adopted."

L245 same comment as for L236
Sentence rewritten. ".. from the neighbouring zone with the most similar tectonic characteristics were adopted."

L361 Figure 4. One would expect also to see (with comments in the text) in this Figure some of the main propminent and known faults in Slovenia(e.g. Idrija, Snežnik, Sava, Orlek) and neighbouting Croatia (e.g. Petrinja). I strongly invite the Authors to add these.
The depth distribution of earthquakes for individual FS was based on the IR18 catalogue compiled from instrumentally recorded earthquakes in the period 1990 – 2018 from seismic catalogues of Slovenia and neighbouring countries. It shows an uneven distribution of earthquakes either because of spatially heterogeneous earthquake activity rates or because of different catalogue's origins (different threshold magnitude/intensity, seismic network density). The best covered areas are Slovenia and Friuli, therefore, depth evaluation using IR18 in these two areas is better than in other areas. For most FS in Croatia and Austria the geological lower depth estimates were chosen.
Histograms and text for some more prominent faults with larger slip rates (Idrija, Raša; Sava E, Labot S) were added to the manuscript and to Fig4.

L379-80 + Fig. 7: »hypocentral depths« Unclear and could not find explanation in the text about the exact definition. Is it the average or median hypocentral depth of the events in the source area or fault related? Same for caption of Figure 7.
Added in line 190: "For AS and PS, the OpenQuake performs calculations considering finite ruptures (GEM, 2022). Therefore, the expected hypocentral depth that represents the centre of finite ruptures, should also be given. It was estimated from measures of central tendency using seismological data."
We changed the term "hypocentral depth" to "expected hypocentral depth" on several places in the document.

L448-451 »... is justified and choice of method appropriate.« Please justify this affirmation with some explanation. Any conclusion on couplings or β parameter? Same applies to Fig. 9. »The fit is the best between..« -->»The best fit is between...«BUT: explain why!
»'Carafa c' branch« Some short explanation about the Carafa branches is needed for the non-specialized reader.
The section covering all three comments was rewritten (now lines 460-472):
"We compared seismologically determined seismic moment rates $\dot{M}\_seis$ for different threshold magnitudes $m_t$ and year of completeness of catalogues with tectonic moment rates $\dot{M}\_tect$. The results (Tables 7 and 8 and Fig. 8) show that the two estimates are not equivalent, possibly due to some aseismic deformation occurring on modelled active faults and wrongly attributed to the seismic deformation if c=1. In such circumstances, additional calculations are needed to remove the active faults' non-seismogenic slip rate. The moment-frequency distribution parameter β relates the seismic moment logarithm to the moment magnitude and to the Gutenberg-Richter b-value ($\beta=2/3b$) (Gutenberg&Richter, 1944; IASPEI, 2013; Kagan, 2002a). Assuming the b-value is one (Šket-Motnikar et al., 2022), $\beta=2/3$. Statistical analysis for moderate earthquakes (Kagan, 2002a) suggests that the β value is 0.60–0.65. In our study area, for β around 0.65, there is an excellent agreement between the seismic moment rate obtained for $m_t$ 4.5 (yellow

dots in Fig. 8) and the seismic moment rate calculated with the seismic coupling (mean estimate, red lines in Fig. 8) as in Carafa et al. (2017).
The good fit between these two alternative estimates indicates the appropriateness of considering an aseismic fraction on the total slip rate for active faults. Also, we assign a higher weight (70 %) to the branch determined using the Carafa et al. (2017) approach, complementary to the branch that corresponds to the fixed c = 0.7 (weight 30 %)."

L598 In caption of Table 9 indicate »SR=slip rate«
Changed text in Table 9 to "slip rate".

L591».... was used as one logic tree branch in PSHA.« ».... was used by (Reference!) as one logic tree branch in PSHA.«
Rewritten.

TECHNICAL COMMENTS All comments are considered in the paper.
L84English: ...are in detail explained --> are explained in detail
L95English:A fraction -->The fraction
L99were studied --> are studied
L166'joined'better 'added'
L172A seismic source ..

**List of relevant changes**

Sentences added to the Abstract:

Line 16
"The hazard model was based on three seismic source models: area source model, fault source model and smoothed seismicity (point) source model."
Line 30
"The presented approach for seismic coupling estimation can be applied in cases where the total slip rate is given instead of its seismic part and can be used at regional or national level. The approach is also suitable for the cross-border harmonisation of the European seismic hazard modeling data. "

Line 80 Sentence changed.
"The development of the Slovenian hazard model ran parallel to the update of the European seismic hazard model (ESHM20) project (Danciu et al., 2021)."

Added sentence in line 89. "Point sources (PS) are centres of grid cells with 10 x 10 km dimensions, which cover the whole influence area of the hazard calculation (Šket Motnikar et al., 2022)."

Added sentence to Fig2 caption." The list of fault source acronyms, names and basic tectonic characteristics are given in the supplement Table S1, the full parametrisation for all fault and area seismic sources is available at https://doi.pangaea.de/10.1594/PANGAEA.940100."
We also added the fault and area source names besides acronyms to histograms in Fig4 and Fig6.

Line 243 Sentence rewritten. ".. from the neighbouring zone with the most similar tectonic characteristics were adopted."

Line 252 Sentence rewritten. ".. from the neighbouring zone with the most similar tectonic characteristics were adopted."

Histograms and text for some more prominent faults with larger slip rates (Idrija, Raša; Sava E, Labot S) were added to the manuscript to chapter 3.1.1 and to Fig4. The names of area source zones were added to histograms in Fig6.

Added in line 191: "For AS and PS, the OpenQuake performs calculations considering finite ruptures (GEM, 2022). Therefore, the expected hypocentral depth that represents the centre of finite ruptures, should also be given. It was estimated from measures of central tendency using seismological data."

We changed the term "hypocentral depth" to "expected hypocentral depth" in several places in the document.

The section 3.2.1 was rewritten (now lines 458-470):
"We compared seismologically determined seismic moment rates $\dot{M}_{seis}$ for different threshold magnitudes $m_t$ and year of completeness of catalogues with tectonic moment rates $\dot{M}_{tect}$. The results (Tables 7 and 8 and Fig. 8) show that the two estimates are not equivalent, possibly due to some aseismic deformation occurring on modelled active faults and wrongly attributed to the seismic deformation if c=1. In such

circumstances, additional calculations are needed to remove the active faults' non-seismogenic slip rate. The moment-frequency distribution parameter β relates the seismic moment logarithm to the moment magnitude and to the Gutenberg-Richter b-value (β=2/3b) (Gutenberg&Richter, 1944; IASPEI, 2013; Kagan, 2002a). Assuming the b-value is one (Šket-Motnikar et al., 2022), $\beta$=2/3. Statistical analysis for moderate earthquakes (Kagan, 2002a) suggests that the β value is 0.60–0.65. In our study area, for β around 0.65, there is an excellent agreement between the seismic moment rate obtained for $m_t$ 4.5 (yellow dots in Fig. 8) and the seismic moment rate calculated with the seismic coupling (mean estimate, red lines in Fig. 8) as in Carafa et al. (2017).

The good fit between these two alternative estimates indicates the appropriateness of considering an aseismic fraction on the total slip rate for active faults. Also, we assign a higher weight (70 %) to the branch determined using the Carafa et al. (2017) approach, complementary to the branch that corresponds to the fixed c = 0.7 (weight 30 %)."

Text was added to Conclusion (lines 620-626)
"Our results contribute to a better understanding of the seismic activity in the region. They also indicate some areas, where future analysis and estimation of slip rates and seismic coupling should be performed more in detail. The presented approach for seismic coupling estimation can be applied in cases where the total slip rate is given instead of its seismic part and can be used in similar tectonic environments. In EFSM20 the seismic coupling is not assigned to individual faults in the dataset. For the moment rate calculations it is conservatively assumed to be equal to 1 and it is thus left to the user to choose a value to apply in applications (Basili et al., 2023). The approach described in this paper is also suitable for applying to such datasets at a regional or national level and therefore enables cross-border harmonisation."

The list of abbreviations is added to the end of the manuscript (lines 628-649).